# KAN SEE YOUR FACE

## ABSTRACT

With the advancement of face reconstruction (FR) systems, privacy-preserving face recognition (PPFR) has gained popularity for its secure face recognition, enhanced facial privacy protection, and robustness to various attacks. Besides, specific models and algorithms are proposed for face embedding protection by mapping embeddings to a secure space. However, there is a lack of studies on investigating and evaluating the possibility of extracting face images from embeddings of those systems, especially for PPFR. In this work, we introduce the first approach to exploit Kolmogorov-Arnold Network (KAN) for conducting embedding-to-face attacks against state-of-the-art (SOTA) FR and PPFR systems. Face embedding mapping (FEM) models are proposed to learn the distribution mapping relation between the embeddings from the initial domain and target domain. In comparison with Multi-Layer Perceptrons (MLP), we provide two variants, FEM-KAN and FEM-MLP, for efficient non-linear embedding-to-embedding mapping in order to reconstruct realistic face images from the corresponding face embedding. To verify our methods, we conduct extensive experiments with various PPFR and FR models. We also measure reconstructed face images with different metrics to evaluate the image quality. Through comprehensive experiments, we demonstrate the effectiveness of FEMs in accurate embedding mapping and face reconstruction.

## 1 INTRODUCTION

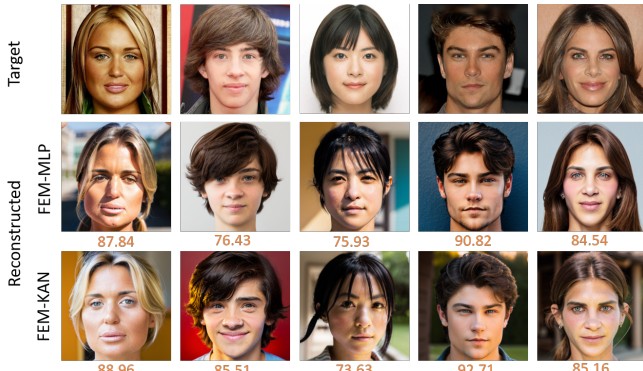

Figure 1: Sample face images from the CelebA-HQ dataset (first row) and their corresponding reconstructed face images from face templates of PPFR model DCTDP. The orange color value indicates confidence score (higher is better) given by commercial API Face++.

The progress of artificial intelligence has brought attention to the security and privacy concerns associated with biometric authentication systems (Laishram et al., 2024; Wang et al., 2024b), specifically face recognition (FR) (Rezgui et al., 2024). FR systems generate the template for each identity for comparing different faces and to authenticate query faces. Those face templates or face embeddings are considered as one type of biometric data that is frequently produced by black-box models (e.g., convolutional neural networks (CNNs) and deep neural networks (DNNs) based models). Existing common threats to the face embeddings are sensitive information retrieve attacks (extract

soft-biometric information such as sex, age, race, etc.) or face reconstruction attacks (recover the complete face image from an embedding). In order to increase the privacy and security level of FR, privacy-preserving face recognition (PPFR) systems (Ji et al., 2022; Mi et al., 2023; Han et al., 2024b;a; Mi et al., 2024) have been proposed. However, most PPFR methods focus on concealing visual information from input face images to the systems but embeddings are not being protected directly. IoM Hashing (Jin et al., 2017) is initially proposed for fingerprint protection by transferring biometric feature vectors into discrete index hashed code. PolyProtect (Hahn & Marcel, 2022) is a mapping algorithm based on multivariate polynomials with user-specific parameters to protect face embeddings. MLP-Hash (Shahreza et al., 2023) proposes a new cancelable face embedding protection scheme that includes user-specific randomly-weighted multi-layer perceptron (MLP) with non-linear activation function and binarizing operation. Homomorphic Encryption (Shahreza et al., 2022b) (HE)-based method is also proposed to encrypt embedding into ciphertext for protection.

Current face reconstruction methods focus on face image reconstruction from embeddings of normal (without special operation for privacy protection on either input face images or face embeddings) FR models. Deconvolutional neural network NbNet (Mai et al., 2018) utilizes the deconvolution to reconstruct face images from deep templates. End-to-end CNN-based method (Shahreza et al., 2022a) combines cascaded convolutional layers and deconvolutional layers to improve reconstruction. Moreover, the learning-based method (Shahreza & Marcel, 2024b) can reconstruct the underlying face image from a protected embedding that is protected by template protection mechanisms (Jin et al., 2004; Shahreza et al., 2023; 2022b). Nevertheless, the reconstructed faces from those methods suffer from noisy and blurry artifacts, which degrade the image naturalness. Generative adversarial network (GAN)-based approach (Otroshi Shahreza & Marcel, 2024) trains a mapping network to transfer face embedding to the latent space of a pre-trained face generation network. However, they only test their method on normal FR systems.

Considering the above motivations, we propose an embedding mapping based face reconstruction framework to generate realistic face images from leaked face embeddings both from normal FR and PPFR models by utilizing a pre-trained IPA-FaceID (Ye et al., 2023) diffusion model. As depicted in Figure 2, we feed training face images to both IPA-FR (default FR of IPA-FaceID) and target FR models. The initial output face embedding from the target FR model is transferred by the Face Embedding Mapping (FEM) model before performing multi-term loss optimization. During the inference stage, the leaked embedding from the target FR model can be mapped by trained FEM and directly used by IPA-FaceID to generate realistic face images. We verify the effectiveness of face reconstruction by applying impersonation attacks to real-world FR systems. Besides, we also provide a test demonstration of FEMs by a commercial face comparison API like Face++[1] as shown in Figure 1.

Our key contributions are:

- We propose a face embedding mapping approach called FEM to map the arbitrary embedding to the target embedding domain for realistic face reconstruction. The trained FEM can be easily integrated into the current SOTA pre-trained IPA-FaceID diffusion model and enable IPA-FaceID generalized for accurate face generation on various types of face embedding, including complete, partial and protected ones.

- To the best of our knowledge, we are the first to exploit the potential of KAN for face embedding mapping and face reconstruction. Compared to the MLP-based model, we showcase the efficacy of the FEM-KAN model for non-linear mapping.

- In contrast to existing face reconstruction methods that aim to inverse face template from normal FR models, we explore the possibility to reconstruct face image from PPFR models. Besides, we conduct extensive experiments in several practical scenarios to test the effectiveness, generalization, robustness and bias of our method. Moreover, we show proposed FEMs can also effectively extract underlying face images from the partial leaked embedding as well as the protected embedding.

---

[1]https://www.faceplusplus.com

## 2 Related Work

### 2.1 ID-Preserving Text-to-image Diffusion Models

Existing text-to-image (T2I) models still have limitation to generate accurate and realistic detailed image due to the limited information expressed by text prompts. Stable unCLIP[2] is based on fine-tuning CLIP image embedding on a pre-trained T2I model to improve the desired image generation ability. IP-Adapter (Ye et al., 2023) proposed decoupled cross-attention to embed image feature from image prompt to a pre-trained T2I diffusion model by adding a new cross-attention layer for image feature, which is separated from text feature. IP-Adapter utilizes trainable projection model to map the image embedding that extracted by a pre-trained CLIP image encoder model (Radford et al., 2021) to a sequence image feature. Later on, IPA-FaceID[3] is developed for customized face image generation by integrating face information through face embedding extracted from a FR model instead of CLIP image embedding. Furthermore, LoRA is utilized to enhance ID consistency. The IPA-FaceID has the ability to produce corresponding diverse styles of image based on a given face and text prompts. Instead of using the pre-trained CLIP model to extract image features, Instant-tID (Wang et al., 2024a) propose a trainable lightweight module for transferring face features from the frozen face encoder into the same space of the text token. Moreover, the IndentityNet based on modified ControlNet (Zhang et al., 2023) is introduced to extract semantic face information from the reference image and face embedding is used as condition in cross-attention layers. ID-conditioned face model Arc2Face (Papantoniou et al., 2024) is based on the pre-trained Stable Diffusion model dedicated for ID-to-face generation by using only ID embedding. It fixes the text prompt with a frozen pseudo-prompt "a photo of $\langle id \rangle$ person" where placeholder $\langle id \rangle$ token embedding is replaced by ArcFace embedding of image prompt. Then the whole token embedding is projected by CLIP encoder to the CLIP output space for training.

### 2.2 From Deep Face Embeddings to Face images

Extracting images from deep face embeddings are challenging for naive deep learning networks e.g., UNet (Ronneberger et al., 2015). (Shahreza et al., 2022a) introduced a CNN-based network to reconstruct face images from corresponding face embeddings that were extracted from the FR model by end-to-end training. With more restrictions on the embedding leakage of FR models, (Shahreza & Marcel, 2024a) attempted to reconstruct the underlying face image from partial leaked face embeddings. They used the similar face reconstruction network in (Shahreza et al., 2022a). However, the reconstructed face images from those two methods are highly blurred. Furthermore, (Shahreza et al., 2024) proposed a new block called DSCasConv (cascaded convolutions and skip connections) to reduce the blurring. However, it still has noticeable blurry artifact around face contour. For more realistic face reconstruction, (Otroshi Shahreza & Marcel, 2024) took the advantage of GAN model to generate face image from the deep face embedding. They employed the pre-trained StyleGAN3 (Karras et al., 2021) network to establish a mapping from facial embeddings to the intermediate latent space of StyleGAN. They constructed the mapping network as two fully connected layers with Leaky ReLU activation function.

## 3 Proposed Method

### 3.1 Kolmogorov-Arnold Theorem Preliminaries

The Kolmogorov-Arnold theorem (Liu et al., 2024) states that any continuous function may be expressed as a combination of a finite number of continuous univariate functions. For every continuous function $f(x)$ defined in the n-dimensional real space, where $x = (x_1, x_2, ..., x_n)$, it can be represented as a combination of a univariate continuous function $\Phi$ and a sequence of continuous bivariate functions $x_i$ and $\phi_{q,i}$. The theorem demonstrates the existence of such a representation:

$$f(x) = \sum_q \Phi_q(\sum_i \phi_{q,i}(x_i)) \tag{1}$$

---

[2]https://huggingface.co/stabilityai/stable-diffusion-2-1-unclip
[3]https://huggingface.co/h94/IP-Adapter-FaceID

This representation suggests that even sophisticated functions in high-dimensional spaces can be reconstructed through a sequence of lower-dimensional function operations.

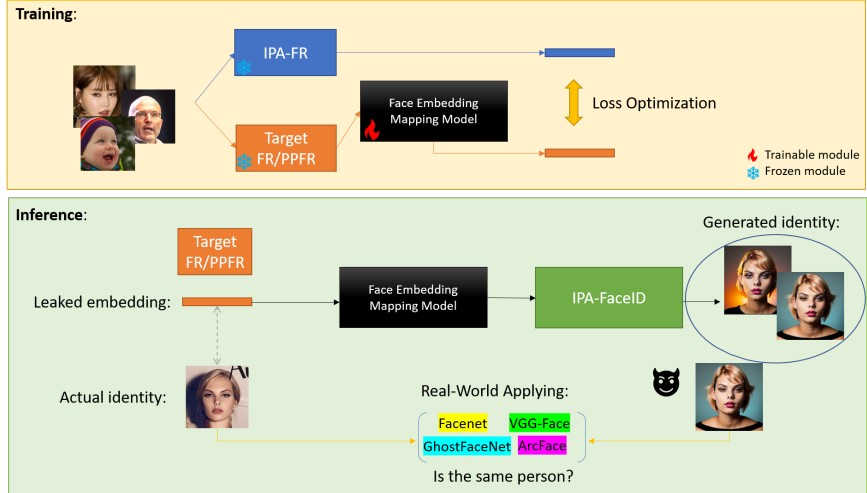

Figure 2: Pipeline of face reconstruction by face embedding mapping.

## 3.2 TRAINING DATA

For training our face reconstruction framework, face embeddings of different identities are needed. Considering the target FR or PPFR model $\Gamma'(.)$ and default FR $\Gamma(.)$ model from IPA-FaceID, for training image dataset $\mathcal{I} = I_i$, we can generate the embedding distribution $\mathcal{D}(e_i)$ as well as $\mathcal{D}'(e_i')$ by extracting face embeddings from all face images in $\mathcal{I}$, where $e_i$ and $e_i'$ denote the output face embeddings from $\Gamma(.)$ and $\Gamma'(.)$.

## 3.3 JOINT LOSS

In order to enable target $\Gamma'(.)$ model to generate realistic target identity face images from IPA-FaceID, the target embedding extracted from $\Gamma'(.)$ should be close to the corresponding embedding that represents the same face identity. Therefore, we should minimize the distance between $\hat{\mathcal{D}}(\hat{e}_i) = \mathcal{M}(\mathcal{D}'(e_i'))$ and $\mathcal{D}(e_i)$, where $\mathcal{M}(.)$ and $\hat{e}_i$ denote FEM and mapped face embedding, respectively.

- Mean Square Error (MSE): To reduce reconstruction difference of the generated embedding, we use MES loss to minimize the square of the reconstruction error:

$$\mathcal{L}_{\text{MSE}}(e_i, \hat{e}_i) = \frac{\sum_{i=0}^{N-1}(e_i - \hat{e}_i)^2}{N} \tag{2}$$

- Pairwise Distance (PD): When p=2, PD computes the pairwise distance between input vectors using the euclidean distance:

$$\mathcal{L}_{\text{PD}}(e_i, \hat{e}_i) = \|e_i - \hat{e}_i\|_p \tag{3}$$

- Cosine Embedding Distance (CED): CED is used for measuring whether two embedding vectors are similar, it is widely used for comparing face template in FR tasks:

$$\mathcal{L}_{\text{CED}}(e_i, \hat{e}_i) = 1 - \cos(e_i, \hat{e}_i) \tag{4}$$

Our total loss is determined by a linear combination of the aforementioned loss types:

$$\mathcal{L}_{\text{total}} = \lambda_1 \mathcal{L}_{\text{MSE}} + \lambda_2 \mathcal{L}_{\text{PD}} + \lambda_3 \mathcal{L}_{\text{CED}} \tag{5}$$

We empirically determined that the selection of $\lambda_1 = 1$, $\lambda_2 = 0.5$, $\lambda_3 = 10$ ($\lambda$ value should be set to balance the range of different loss functions) yields the best performance. See in Section 5.5 for the performance of different reconstruction loss functions.

## 3.4 FACE EMBEDDING MAPPING (FEM)

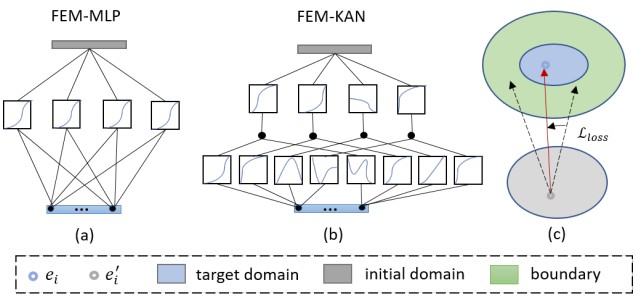

Figure 3: Two variants of FEM models and the process of embedding-to-embedding mapping. (a) FEM-MLP has fixed activation function. (b) FEM-KAN has learnable activation function at edges to achieve accurate non-linear mapping. (c) The direction of embedding mapping optimized by distance towards to 'ground truth' face embedding $e_i$.

Face embedding is a vector that represents facial information associated with a corresponding identity. Ideally, embeddings that are extracted from different face images of the same identity should be close and far for those that computed from different ones. Existing SOTA FR and PPFR networks utilize similar structures of backbone to extract features from the face image and compute the face template or face embedding. We assume there is a transformation or mapping algorithm between embeddings from the same identity that are extracted by different backbones. Inspired from (Papantoniou et al., 2024) and (Liu et al., 2024), we propose FEM-MLP and FEM-KAN showing in Figure 3 to learn the mapping relation of embedding distributions from different FR backbones. Then trained FEMs can map face embedding from the initial domain into the corresponding target domain of the pre-trained IPA-FaceID diffusion model in order to generate face images. Depending on the effectiveness of FEMs, the mapped embedding can fall into the target domain and boundary region. The target domain represents mapped embedding can be used for ID-preserving face image generation that can fool the evaluation FR systems while boundary region indicates mapped embedding is not sufficient for ID-preserving face image generation but human-like image generation.

## 4 EXPERIMENTS

### 4.1 EXPERIMENTAL DETAILS

To evaluate the reconstruction performance of the proposed face reconstruction network, we train the two variant FEM models on various target SOTA FR and PPFR models. We generate 5000 images as training dataset based on the subset of CelebA-HQ (Karras, 2017) by applying Arc2Face (Papantoniou et al., 2024) model, five images generated for each identity. FR `buffalo_l`[4] is selected as the defualt FR model of IPA-FaceID. We choose `faceid_sd15`[5] checkpoint for IPA-FaceID which takes face embedding and text as input. In order to effectively generate face in proper angle, we fix the text prompt as "front portrait of a person" for all the experiments. We follow `efficient_kan`[6] for FEM-KAN implementation and set the same hidden layer structure *[512, 1024, 3072, 512]* for PEM-MLP. We use the GELU activation function and add 1D batch normalization to FEM-MLP.

Our goal is to reconstruct complete face image from embedding of target PPFR models. Then, the generated face images are used to access or fool other FR systems in order to verify the reconstruction performance. We conduct mainly four different experiments to verify the proposed method as follows:

- To show the effectiveness of FEMs, we reconstruct five face images for each embedding that extracted from target models and inject the generated faces to the evaluation FR models for performing face verification.

---

[4]https://github.com/deepinsight/insightface
[5]https://huggingface.co/h94/IP-Adapter-FaceID/tree/main
[6]https://github.com/Blealtan/efficient-kan

- In order to test the generalization of FEMs, we train models on 90% Flickr-Faces-HQ (FFHQ) (Karras et al., 2019) dataset and test on customized dataset Synth-500, with 500 images of never-before-seen identities, generated by website[7].

- For evaluating the robustness of FEMs, we consider reconstructing face from partial embeddings instead of complete ones. We set different levels of embedding leakage starting from 10% to 90% (e.g., removing the last 10% values from each embedding vector). Except, we also conduct experiments for reconstructing faces from the protected embedding, e.g., PolyProtect and MLP-Hash.

- For evaluating the bias of face reconstruction to identity from different demographic, we test our method on Racial Faces in-the-Wild (RFW) dataset which includes four races such as Caucasian, Asian, Indian and African. We select 10% of each race in our experiment by considering the testing time, each image is loosely cropped into size $112{\times}112$.

Our experiments are conducted on a Tesla V100 GPU with 32G memory using PyTorch framework, setting the batch size to 128. For optimizers, we use SGD and AdamW for FEM-KAN and FEM-MLP with initial learning rate $10^{-2}$ and $10^{-3}$, the exponential learning rate decay is set to 0.8 for AdamW.

### 4.2 TARGET MODELS AND REAL-WORLD MODELS

For target models that we aim to reconstruct face image from can be categorized into normal FR models such as IRSE50 (Hu et al., 2018), IR152 (Deng et al., 2019) and PPFR models including DCTDP (Ji et al., 2022), HFCF (Han et al., 2024b), HFCF-SkinColor (Han et al., 2024a) and PartialFace (Mi et al., 2023). All face embeddings extracted by FR and PPFR models have the same length equal to 512. We have selected four widely-used public FR models as real-word models for face verification in order to test performance of face reconstruction. These models are FaceNet (Schroff et al., 2015), VGG-Face (Parkhi et al., 2015), GhostFaceNet (Alansari et al., 2023), ArcFace (Deng et al., 2019) and we use implementation from `deepface`[8].

### 4.3 EVALUATION METRICS

For evaluation, we employ the attack success rate (ASR) as a metric to assess the attack efficacy to various target FR systems by reconstructed face images from target PPFR systems. ASR is defined as the fraction of generated faces successfully classified by the target FR system. We generate five images for each embedding. When determining the ASR, we establish a False Acceptance Rate (FAR) of 0.01 for each FR model. Furthermore, we employ FID, PSNR, SSIM, LPIPS and maximum mean discrepancy (MMD) (Borgwardt et al., 2006) to evaluate the image quality of generated face images. SSIM and LPIPS are metrics based on perception, while MMD has the ability of analyzing two data distributions and assessing the imperceptibility of the generated images (Yang et al., 2021).

## 5 RESULTS

### 5.1 PERFORMANCE ON BLACK-BOX ATTACKING

As shown in Table 1, our proposed face reconstruction network can effectively extract face images from target models. Comparing the baseline case (without applying any embedding mapping algorithms), our method substantially increases the ASR e.g., about 58.3% and 62.9% on FEM-MLP and FEM-KAN method in average for target DCTDP model. Among all the target PPFR models, DCTDP is less robust against face reconstruction attack, where FEM-KAN achieves 67.6%. It shows that even images transferred into frequency domain, the corresponding face embedding still contain information can be used for face reconstruction. FEM-KAN has overall better performance than MLP-based FEM in general, especially for target PPFR models. The potential reason is that the embedding distribution from PPFR models is more far away from the idea distribution that can

---

[7]https://thispersondoesnotexist.com
[8]https://github.com/serengil/deepface

Table 1: Evaluations of Attack Success Rate (ASR) for black-box attacks to FR and PPFR models on CelebA-HQ dataset. Other four FR models are used for verifying the efficacy of FEMs.

| Target Model | Method | Facenet | VGG-Face | GhostFaceNet | ArcFace | Average |
|---|---|---|---|---|---|---|
| | None | 3.9 | 6.3 | 3.0 | 9.1 | 5.6 |
| IRSE50 (Hu et al., 2018) | MLP | 33.8 | 63.4 | 64.2 | 72.8 | 58.6 |
| | KAN | 25.3 | 53.7 | 75.2 | 67.9 | 55.5 |
| | None | 4.6 | 4.0 | 7.6 | 2.3 | 4.6 |
| IR152 (Deng et al., 2019) | MLP | 39.2 | 58.7 | 68.0 | 78.9 | 61.2 |
| | KAN | 36.7 | 56.3 | 68.8 | 80.7 | 60.6 |
| | None | 3.8 | 5.0 | 2.9 | 7.1 | 4.7 |
| DCTDP (Ji et al., 2022) | MLP | 39.5 | 64.3 | 69.8 | 78.4 | 63.0 |
| | KAN | 45.2 | 67.4 | 75.2 | 82.7 | 67.6 |
| | None | 6.5 | 6.9 | 5.1 | 11.4 | 7.5 |
| HFCF (Han et al., 2024b) | MLP | 34.0 | 58.7 | 63.2 | 74.9 | 57.7 |
| | KAN | 38.5 | 62.9 | 70.8 | 77.7 | 62.5 |
| | None | 3.3 | 4.8 | 1.9 | 7.2 | 4.3 |
| HFCF-SkinColor (Han et al., 2024a) | MLP | 35.2 | 61.8 | 62.4 | 76.6 | 59.0 |
| | KAN | 42.0 | 67 | 69.0 | 81.6 | 64.9 |
| | None | 2.5 | 3.5 | 1.8 | 6.6 | 3.6 |
| PartialFace (Mi et al., 2023) | MLP | 35.7 | 59.4 | 63.0 | 72.6 | 57.7 |
| | KAN | 39.4 | 64.4 | 68.4 | 76.0 | 62.1 |

be utilized by IPA-FaceID. Therefore, simple model like MLP can not perfectly map the distribution. Among the target PPFR models, FEMs have the lowest ASR on PartialFace with average ASR 57.7% and 62.1% on FEM-MLP and FEM-KAN.

The embedding mapping and face reconstruction performance are associated with the capability of feature extractor. In order to study the impact of complexity of feature extractor to the face reconstruction, we trained PPFR PartialFace with two different backbones.

Table 2: Effects of the backbone to Attack Success Rate (ASR) on PPFR PartialFace (Mi et al., 2023).

| Backbone | Method | Facenet | VGG-Face | Ghost-FaceNet | ArcFace | Average |
|---|---|---|---|---|---|---|
| | None | 3.5 | 5.1 | 2.1 | 10.3 | 5.3 |
| ResNet18 | FEM-MLP | 14.3 | 35.0 | 29.1 | 43.2 | 30.4 |
| | FEM-KAN | 14.5 | 39.4 | 31.9 | 45.7 | 32.9 |
| | None | 2.5 | 3.5 | 1.8 | 6.6 | 3.6 |
| ResNet34 | FEM-MLP | 35.7 | 59.4 | 63.0 | 72.6 | 57.7 |
| | FEM-KAN | 39.4 | 64.4 | 68.4 | 76.0 | 62.1 |

Table 2 demonstrates that the incorporation of a substantial number of layers in the backbone of the PPFR model results in superior performance of FEMs regarding ASR. The deep backbone can derive more identity-consistent information from intra-class images. Consequently, the mapping relationship between inter-class identities is relatively straightforward for FEMs to learn.

## 5.2 PERFORMANCE ON IMAGE QUALITY

Table 3: Quantitative evaluations of image quality on Synth-500 dataset. The target model is ArcFace. FEMs are trained on FFHQ dataset.

| Method | FID $\downarrow$ | PSNR $\uparrow$ | SSIM $\uparrow$ | MMD $\downarrow$ | LPIPS $\downarrow$ |
|---|---|---|---|---|---|
| None | 179.4421 | 8.0349 | 0.32411 | 34.6578 | 0.5302 |
| FEM-MLP | 89.1635 | 11.5839 | 0.43318 | 33.1191 | 0.4265 |
| FEM-KAN | **72.7869** | **11.8401** | **0.43524** | **33.1080** | **0.4156** |

We report image quality assessment result on images that generated from face embedding that extracted from never-before-seen identities. In Table 3, FEM-KAN has better performance on all four

different metrics and it shows the generalization ability of FEM-KAN to generate high quality face images from new identities. Since MMD requires 1D input, we flatten image into 1D vector before calculation. Due to the gender uncontrollable face reconstruction, there is still space for improvement in terms of image quality, e.g., FID. Another important observation is that the image quality metrics might not fairly reflect the effectiveness of generated face images since they are not align with the visual similarity. See more discussion in Appendix A.4.

As shown in Figure 4, we plot embedding similarity distributions on whole dataset Synth-500, ArcFace model (clean) has very poor ability to extract the proper embedding that can be used for face generation with mean cosine similarity around 0.1357. The generated face images from clean ArcFace model barely can be used for accessing other FR models regarding the distribution of cosine similarity. In contrast, by mapping the embedding from FEMs, the cosine similarity gets increased significantly, and the FEM-KAN has relatively more generated images with high similarity than FEM-MLP. According to the identity similarity distributions after applying FEMs, we can see the majority of failed reconstructed samples with cosine similarity around 0.1 while only limited samples have been perfectly reconstructed with cosine similarity around 0.9.

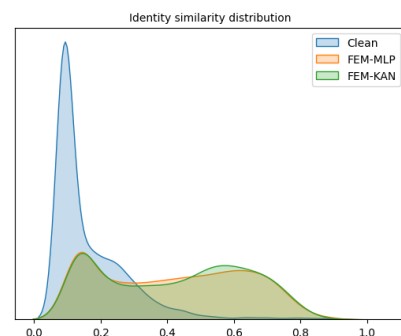

Figure 4: Cosine similarity distributions between input and generated faces from FEMs. ArcFace is used as target model to extract embeddings from Synth-500. FEMs are trained on FFHQ dataset.

## 5.3 FACE RECONSTRUCTION FROM PARTIAL LEAKED EMBEDDINGS

Table 4: Attack Success Rate (ASR) on different percentage of embedding leakage. The target model is IRSE50 with evaluation on ArcFace model.

| Method | 10% | 30% | 50% | 70% | 90% |
|---------|------|------|------|------|------|
| FEM-MLP | **15.2** | **31.2** | **50.1** | **61.4** | **69.9** |
| FEM-KAN | 14.5 | 21.5 | 40.6 | 57.6 | 68.0 |

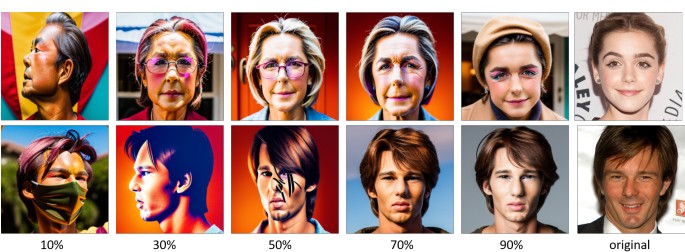

Figure 5: Reconstructed faces by FEM-KAN from different percentage of embedding leakage. IRSE50 is target model.

Previous experiments are based on assumption that adversary can gain access to the complete face embeddings. Nevertheless, in some real-world scenarios, a complete face embedding is difficult to acquire, but rather to access a portion of the embedding. For example, face embeddings of the FR system are split and stored on different servers for data protection like the situation considered in (Shahreza & Marcel, 2024a). We assume adversary already trained FEMs on complete embeddings of target FR or PPFR model. In order to further test the FEMs non-linear mapping ability and face reconstruction, we only use partial leaked embeddings (e.g., discarding the second half of values in an embedding in case of 50% leakage) as input to trained FEMs. In order to match the input shape requirement of FEMs, we append zeros to the end of each leaked embedding vector to make the embedding have a length equal to 512.

Table 4 reports ASR to evaluate incomplete leaked embedding mapping ability of FEMs. With increased percentage of embedding leakage, the number of generated face images that can fool the evaluation FR is reduced. ArcFace FR system is configured at FMR of 0.1%. PEM-KAN is able to maintain the same face reconstruction performance by using 90% of embedding compared with complete embedding. As for 70% leakage, model still can achieve relatively high ASR. Figure 5 depicts sample face images from the CelebA-HQ dataset and the corresponding reconstructed face images from partial embedding with different leakage percentages. The reconstructed face images can reveal privacy-sensitive information about the underlying user, such as gender, race, etc. However, the generated face tend to have noticeable artifacts when leakage is lower than 50%. Consequently, we raise the issue of face embedding security in the partial leakage scenario.

## 5.4 FACE RECONSTRUCTION FROM PROTECTED EMBEDDINGS

Considering more strict to the accessing original embeddings that directly computed by the feature extractor of PPFR models, we test FEMs on face embeddings that being protected by particular embedding protection algorithms such as PolyProtect (Hahn & Marcel, 2022) and MLP-Hash (Shahreza et al., 2023). We train FEMs directly on protected face embeddings from both protection algorithms. For PolyProtect, we generate the user-specific pair for each identity in the testing dataset. After mapping original face embedding from PPFR model, the protected embedding from PolyProtect has reduced dimension, 508 in our setting. During training, we append other four zeros to end of protected embedding to maintain the length of vector. For MLP-Hash method, we set one-hidden layer with 512 neurons and fix the seed for all identities. More details about PolyProtect and MLP-Hash are in Appendix A.3.

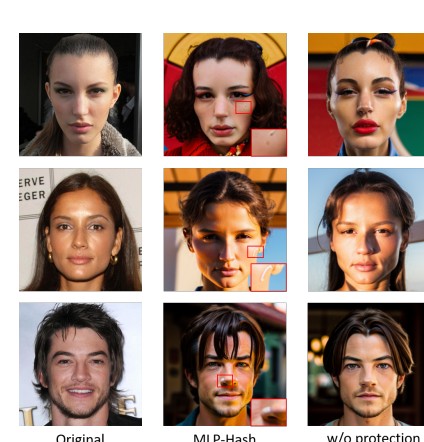

Original      MLP-Hash      w/o protection

Figure 6: Reconstructed faces from protected embeddings.

Table 5: Attack Success Rate (ASR) performance on protected face embeddings. HFCF is target model.

| Protection Algorithm | Method | Facenet | VGG-Face | GhostFaceNet | ArcFace |
|---|---|---|---|---|---|
| PolyProtect | FEM-MLP | 5.0 | 9.3 | 6.8 | **15.6** |
| | FEM-KAN | **11.2** | **10.7** | **8.7** | 14.5 |
| MLP-Hash | FEM-MLP | 23.4 | 47.3 | 51.9 | 64.5 |
| | FEM-KAN | **25.3** | **53.0** | **56.5** | **71.6** |

Table 5 reports the ASR of on the embeddings protected by PolyProtect and MLP-Hash. It is worth to notice that FEMs achieve high face reconstruction performance against MLP-Hash and have comparable ASR with the ones on unprotected embeddings in Table 1. Moreover, FEM-KAN has higher ASR in all the evaluation FR models than FEM-MLP which indicates KAN's superiority in terms of learning the non-linear relation. However, FEMs are not able to effectively extract underlying faces from protected embeddings of PolyProtect. The extreme large and small values in protected embeddings after mapping by PolyProtect might make FMEs difficult to learn. As showing in Figure 6, reconstructed faces from embeddings protected by MLP-Hash tend to have certain artifact within the face. The potential reason can be the limited information presented in binarized face embeddings after applying MLP-Hash.

## 5.5 ABLATION STUDY

**Effects of different loss functions.** To evaluate the impact of loss function to face reconstruction, we test the three loss function configurations with IR50 FR model. As showing in Table 6, we train FEM-KAN for 20 epochs on each loss function setting. It is worth to notice that $\mathcal{L}_{\mathrm{MSE}}$ term greatly

Table 6: Attack Success Rate (ASR) with various reconstruction loss function configurations on CelebA-HQ. IR50 is used as target model here.

| Loss Function | Facenet | VGG-Face | GhostFaceNet | ArcFace | Average |
|---|---|---|---|---|---|
| $\mathcal{L}_{PD}$ | 23.9 | **54.0** | 54.8 | **68.7** | 50.35 |
| $\mathcal{L}_{PD} + \mathcal{L}_{CED}$ | 21.2 | 53.3 | 49.7 | 65.3 | 47.38 |
| $\mathcal{L}_{MSE} + \mathcal{L}_{PD} + \mathcal{L}_{CED}$ | **25.3** | 53.7 | **75.2** | 67.9 | **55.5** |

improve the face image reconstruction performance compared with other two loss terms, especially it increases more than 20% ASR on GhostFaceNet.

**Failed cases and bias.** Although the pretrained IPA-FaceID has ability to generate face image even on "weak" face embedding which is not accurately mapped by FEMs, we found that the reconstruction rate for male is much lower than for the female identity as showing in Figure 7. Such observations may be due to the image generation bias in pre-trained IPA-FaceID. For target PPFR HFCF, FEMs has lowest ASR on African group of RFW dataset, 21.7%, 19.7%, 17.5% lower than Caucasian, Asian and Indian groups as stated in Table 7. Due to the low resolution images of RFW dataset, face reconstruction performance is reduced on every group.

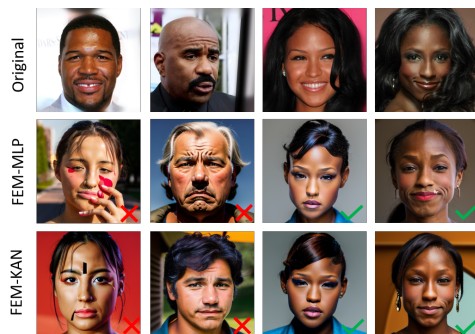

Figure 7: Failed samples from HFCF. The red and green symbol indicate generated face image passed and failed in face verification.

Table 7: Attack Success Rate (ASR) performance on RFW dataset. ArcFace is the evaluation FR.

| Target PPFR | Method | RFW | | | |
|---|---|---|---|---|---|
| | | Caucasian | Asian | Indian | African |
| HFCF (Han et al., 2024b) | FEM-MLP | 51.1 | 52.6 | 48.8 | 33.6 |
| | FEM-KAN | **59.0** | **57.0** | **54.8** | **37.3** |

The bias in face generation can be inherited from the face extractor used for IPA-FaceID. Due to unbalanced and biased training dataset of pre-trained model, FR and PPFR models have different ability (see in Appendix A.2) to extract and recognize faces from various races.

## 6 CONCLUSION

In this paper, we propose a new method to reconstruct diverse high-resolution realistic face images from face embeddings in both FR and PPFR systems. We use a pre-trained IPA-FaceID network and trained the mapping model FEM to transfer the embedding for complete face reconstruction, especially, two variant FEMs are proposed for comparison. We conduct comprehensive experiments covering two datasets to measure the face reconstruction performance in different scenarios including black-box embedding-to-face attacks, out-of-distribution generalization, reconstructing faces from protected embeddings and partial leaked embeddings, and bias studies in face reconstruction. To the best of our knowledge, it is a very first work to invert face embedding from PPFR models to generate realistic face images. Extensive experimental evaluations demonstrate that FEMs can improve the face generation ability of pre-trained IPA-FaceID by a substantial margin on privacy-preserving embeddings of PPFR models. We would like to draw the attention of researchers concerning face embedding protection in scenarios of diffusion models. Due to the limitations of feature extractor and pre-trained IPA-FaceID, our method is less effective to produce low-resolution face images. For the future work, we consider improving the gender-preserving ability of our method and reducing the bias in the image generation.

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

# A APPENDIX

## A.1 PRIVACY-PRESERVING FACE RECOGNITION (PPFR) CONFIGURATIONS

Table 8: Model training configurations.

| Parameters | Value |
|---|---|
| Backbone | ResNet34 (He et al., 2016) |
| Optimizer | SGD |
| Loss Function | ArcFace |
| Epoch | 24 |
| Batch Size | 128 |

For the detailed setting of ArcFace loss, scale $s = 64$, weight $w = 1.0$ and margin $m = 0.3$. Besides, we use default training configurations for PartialFace implementation (Mi et al., 2023)[9], 27 random sub channels are selecting for training. The only two differences are that we use ResNet34 as backbone and VGGFace2 (Cao et al., 2018) dataset for training in order to have the same setting with other PPFR models implementation in our work.

During the training stage of PartialFace, the input RGB image is transferred into the frequency domain by discrete cosine transform (DCT) (Ahmed et al., 1974). The initial number of frequency channels is reduced to 132 from 192 after removing 30 low frequency channels. Then for each identity, 27 channels are selected randomly according to the corresponding label. During the inference

---

[9]https://github.com/Tencent/TFace/tree/master/recognition/tasks/partialface

stage, we consider two different scenarios, adversary has or does not have access to the label of leaked embedding. For the latter case, we randomly generate a number from [0, 1000] as the label of leaked embedding for inference.

Table 9: Effects of different subset of frequency channels to Attack Success Rate (ASR) on PPFR PartialFace (Mi et al., 2023).

| If adversary knows label | Method | Facenet | VGG-Face | Ghost-FaceNet | ArcFace | Average |
|---|---|---|---|---|---|---|
| Yes | FEM-MLP | 35.7 | 59.4 | 63.0 | 72.6 | 57.7 |
|  | FEM-KAN | 39.4 | 64.4 | 68.4 | 76.0 | 62.1 |
| No | FEM-MLP | 35.9 | 58.2 | 67.8 | 73.4 | 58.8 |
|  | FEM-KAN | 38.7 | 64.3 | 67.8 | 75.7 | 61.6 |

As shows in Table 9, FEMs can efficiently mapping the embedding whether with knowledge of label or not.

## A.2 PRIVACY-PRESERVING FACE RECOGNITION (PPFR) BIAS ON RFW DATASET

Table 10: PPFR face verification performance on RFW dataset.

| Target PPFR | RFW | | | |
|---|---|---|---|---|
|  | Caucasian | Asian | Indian | African |
| DCTDP (Ji et al., 2022) | **95.48** | 90.63 | 92.90 | 91.75 |
| HFCF (Han et al., 2024b) | **94.71** | 91.35 | 88.61 | 90.03 |
| HFCF-SkinColor (Han et al., 2024a) | **94.80** | 88.98 | 91.52 | 89.93 |
| PartialFace (Mi et al., 2023) | **94.23** | 89.27 | 90.76 | 87.70 |

As depicted in Table 10, we test PPFR models that used in our work on RFW dataset to show the racial bias. PPFR models have much lower accuracy on non-Caucasians than Caucasians.

## A.3 EMBEDDING PROTECTION ALGORITHM IMPLEMENTATION

For the consistent notation, we denote the original face embedding $V = [v_1, v_2, ..., v_n]$ and protected face embedding as $P = [p_1, p_2, ..., p_n]$.

**PolyProtect implementation.** The mapping operation is achieved by following formula. For the first value in $P$,

$$p_1 = \mathsf{c}_1 * v_1^{\mathsf{e}_1} + \mathsf{c}_2 * v_2^{\mathsf{e}_2} + ... + \mathsf{c}_m * v_m^{\mathsf{e}_m} \tag{6}$$

where $\mathsf{C} = [\mathsf{c}_1, \mathsf{c}_2, ..., \mathsf{c}_m]$ and $\mathsf{E} = [\mathsf{e}_1, \mathsf{e}_2, ..., \mathsf{e}_m]$ are 1D vectors that contain non-zero integer coefficients. Each m consecutive values in $V$ are mapped into the corresponding value in $P$. For the range of E, large numbers should be avoided due to small floating numbers of face embeddings are tended to be to zero when large index number in exponential function. However, the range C selection is arbitrary since the PolyProtect is not affected by amplitude. We keep m = 5, E in the range [1, 5], C in the range [-50, 50] as used in paper Hahn & Marcel (2022). Overlap parameter indicates the number of the same values from $V$ that are selected for calculation of each value in $P$. For detailed information about this parameter, we suggest readers to see the original implementation of PolyProtect (Hahn & Marcel, 2022).

**MLP-hash implementation.** It has two stages in MLP-hash including pseudo-random MLP and Binarizing. In the first stage, the pseudo-random matrix $M_\ell$ within range [0,1] is generated from the uniform distribution according to user specified seed. Gram-Schmidit is applied to each row of $M_\ell$ to compute orthonormal matrix $M_{\perp\ell}$. The protected embedding before binarizing $P$ is calculated as:

$$P = F(V \times M_{\perp\ell}) \tag{7}$$

where $F(.)$ denotes activation function, it is a nonlinear function that converts negative value to zero. The number of MLP hidden layers determines the number of iteration in this stage.

Then the final binarized protected embedding can be computed as:

$$p_i = \begin{cases} 0, & \text{if } v_i \leq \tau \\ 1, & \text{otherwise} \end{cases} \tag{8}$$

For detailed algorithm, see in MLP-hash paper Shahreza et al. (2023).

### A.4 Image Quality Metrics not Revealing Perceptual Similarity

As shown in Table 3, we report image quality mainly using FID, PSNR, SSIM, MMD and LPIPS. However, we only achieve marginally better performance on metrics after applying FEMs. The potential reason is that those metrics are not strongly associated with perceptual similarity.

Sample images of the same identity

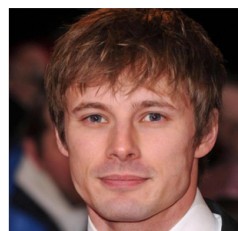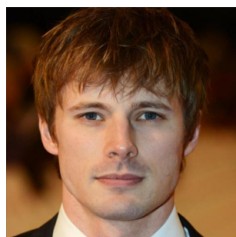

PSNR: 13.8134  LPIPS: 0.4330
SSIM: 0.2182    Cosine Similarity: 0.8013

Figure 8: Image quality metrics are not perfect align with visual similarity. Samples are taken from other subset of CelebA-HQ.

In Figure 8, we select two images from the same person and calculate the corresponding evaluation metrics between them. We exclude FID and MMD metrics since the FID requires multiple images for calculation while the latter one is completely non-relevant with visual similarity as mentioned in paper Borgwardt et al. (2006). We can see the calculated values for PSNR, SSIM and LPIPS all indicate 'low' image quality when considering good result values for PSNR are around 30 to 50 and 0.8 to 1 for SSIM. However, the cosine similarity metric reflects better alignment with visual similarity in this case. Hence, we argue that the image quality metrics might not be the perfect measurement for evaluating the performance of our proposed method. We will consider evaluating our model on some perceptual-related metrics, especially those dedicated for faces (Sadovnik et al., 2018) in future work.

