# OpenReview forum: "KAN See Your Face"
_ICLR.cc/2025/Conference — ICLR 2025 Conference Withdrawn Submission_

### Official Review · Reviewer_BosT · 2024-10-18

**Soundness:** 1
**Presentation:** 1
**Contribution:** 1
**Rating:** 3
**Confidence:** 4

**Summary:**

This paper presents an approach for reconstructing face images from feature embeddings extracted from a face recognition (FR) model, or from a post-processing face recognition (PPFR) system. The authors propose using a network to achieve a transformation in the feature space, mapping these embeddings to the latent space of the IPA-FaceID model, which then generates the corresponding face image.

**Strengths:**

N/A

**Weaknesses:**

I think this paper are not prepared to be submitted since it has some deadly flaws:
- The literature review in this paper is insufficient. There are already many attack methods (refer to [1-4]) for reconstructing faces from face embeddings, but this paper has no comparsion, and even claims that
> To the best of our knowledge, we are the first to exploit the potential of KAN for face embedding mapping and face reconstruction.
- The paper fails to introduce novel insights or methodologies in the field of face reconstruction from embeddings. The methods and techniques presented lack originality and do not significantly advance the state-of-the-art.  This paper merely combines the KAN and IPA-Face with limited adjustment. I can not get the motivation behind it.
- The attack is too naive. This paper only consider white-box attack, with some data available. I recommend the author follow the attack settting in some valueable papers. More attack scenarios should be taken into consideration, please refer to [1][5][6].

[1]Inference Attacks Against Face Recognition Model without Classification Layers[J]. arXiv preprint arXiv:2401.13719, 2024.

[2]Realistic face reconstruction from deep embeddings, NeurIPS 2021 Workshop Privacy in Machine Learning, 2021.

[3]Face Reconstruction from Deep Facial Embeddings using a Convolutional Neural Network, IEEE International Conference on Image Processing (ICIP), 2022

[4]Reconstruct face from features based on a genetic algorithm using GAN generator as a distribution constraint, Computers & Security, vol. 125, p. 103026, Feb. 2023, doi: 10.1016/j.cose.2022.103026.

[5]Mirror: Model inversion for deep learning network with high fidelity. In Proceedings of the 29th Network and Distributed System Security Symposium, 2022

[6] Plug & play attacks: Towards robust and flexible model inversion attacks. In Proceedings of the 39th International Conference on Ma653 chine Learning (ICML)

**Questions:**

Refer to Weaknesses.

---

### Official Review · Reviewer_Um3K · 2024-10-31

**Soundness:** 2
**Presentation:** 2
**Contribution:** 2
**Rating:** 5
**Confidence:** 5

**Summary:**

The paper presents a recovery attack methodology targeting human face recognition embeddings. It introduces a feature conversion module designed for non-linear mapping of embeddings, effectively translating the feature space of various face recognition models into the model space utilized by the pretraining diffusion model. Subsequently, a generative model is employed to reconstruct the facial images. The paper details two structured implementations of the feature conversion modules, named FEM-KAN and FEM-MLP. The efficacy of the feature mapping and face conversion techniques is validated through a comprehensive set of experimental results.

**Strengths:**

（1）The paper leverages an advanced pre-trained diffusion model to execute a face recognition embedding-to-image restoration attack, which proves to be an effective approach. （2）The paper employs a straightforward conversion network to adapt a pre-trained generative model for compatibility with multiple recognition models, which appears to be a simple yet effective strategy.

**Weaknesses:**

（1） While the final experimental outcomes in the paper affirm the method's effectiveness, I have reservations about some of the experimental setups used. Specifically, the paper focuses on the restoration attack of face recognition embeddings, yet employs the PPFR method in its experiments, which is primarily designed to protect the original face image. This setup may not adequately demonstrate the method's effectiveness as described in the paper. Ideally, the method should be tested for its attack capabilities against embedding protection techniques. Although validations on polyprotect and mlp-hash are included, it is crucial to extend these verifications to a broader range of template protection methods, particularly those that differ from the aforementioned, such as methods based on Homomorphic Encryption. （2）The paper presents two conversion structures, FEM-MLP and FEM-KAN, with the integration of Kan into FEM-KAN being emphasized as a key contribution by the authors. The experimental data appears to indicate that FEM-KAN performs better than FEM-MLP. However, as highlighted in lines 284-285, there are inconsistencies in the experimental setups used for each structure. These discrepancies question the reliability of the comparative results and the assertion that FEM-KAN is superior. To solidify this claim, a more comprehensive discussion and detailed analysis are recommended to effectively validate the superiority of FEM-KAN. （3）The baseline result indicated in the "None" row of Table 1 should be reconsidered. The discrepancy between the recognition model used to develop the generative model and the recognition model targeted in the attacks could predictably lead to a diminished attack success rate. A more appropriate baseline would involve fine-tuning the generative model using the same recognition model that is subject to the attack. This adjustment would provide a more accurate and fair comparison of the attack's effectiveness.

**Questions:**

The experimental settings outlined in the paper could benefit from certain adjustments to more effectively validate the proposed method's efficacy.

**Details Of Ethics Concerns:**

It's important to address the ethical considerations regarding the use of 500 pictures from the website, as described in lines 270 to 272 of the paper.

---

### Official Review · Reviewer_NrF1 · 2024-11-01

**Soundness:** 2
**Presentation:** 2
**Contribution:** 2
**Rating:** 1
**Confidence:** 2

**Summary:**

This paper introduces a method for reconstructing realistic face images from face embeddings, including those generated by privacy-preserving face recognition (PPFR) models. Using Face Embedding Mapping (FEM) models, specifically FEM-KAN with Kolmogorov-Arnold Networks and FEM-MLP, the authors map embeddings to a target domain to enable effective face reconstruction through non-linear embedding transformations. Extending existing work, this approach demonstrates the vulnerability of PPFR systems to face reconstruction attacks, showing that identifiable faces can be recovered even from partially leaked or protected embeddings. Extensive experiments validate the model's efficacy across various embedding sources, including normal and privacy-enhanced systems, and evaluate its performance in terms of generalization, robustness, and demographic bias.

**Strengths:**

1, By using Kolmogorov-Arnold Networks (KAN) within the Face Embedding Mapping (FEM) model, it introduces a creative method for embedding-to-embedding mapping that distinguishes it from previous face reconstruction techniques.

2. The authors conduct extensive experiments across a broad range of models, rigorously validating FEM’s robustness with both quantitative and qualitative metrics.

**Weaknesses:**

This paper has several notable weaknesses that impact its contribution, novelty, rigor, and thoroughness.

1. The paper does indeed show a lack of novelty in certain technical aspects. While effectively applied, techniques like KAN, IPA-FaceID, and loss functions are previously established methods that do not advance the foundational technology or introduce significant innovation.

2. The theoretical justification for using KAN as the face embedding mapping model is superficial. The paper does not clearly articulate why KAN offers advantages over MLP in this context, nor does it explain the relevance of KAN’s theoretical underpinnings for mapping embeddings. While experimental results are favorable, the weak theoretical support limits the reader’s understanding of KAN’s suitability for this task.

3. The loss function design appears redundant and lacks a proper explanation. The use of both Pairwise Distance (PD) loss and Cosine Embedding Distance (CED) is redundant, as they become identical when embeddings are normalized, and Mean Square Error (MSE) is also similar to PD. The absence of a clear rationale for these choices raises questions about their necessity, and the ablation study does not effectively isolate the impact of each loss component, making it unclear if all losses contribute meaningfully.

4. The reconstruction attack experiments are not comprehensive. The study would benefit from aligning with a more established protocol, such as the four attack scenarios described by [1], which provide a structured and well-recognized framework for evaluating inversion attacks. Adopting this protocol would enhance the robustness of the attack analysis and allow for easier comparisons with similar studies.

5. The paper overlooks an important class of face template protection: cancellable biometrics. This approach involves transformation-based methods designed to be cancelable upon compromise. They should be included to assess inversion attacks on protected templates.

6. The approach's dependency on pre-trained models like IPA-FaceID raises questions about the reconstruction method’s flexibility and independence from specific architectures. This reliance could restrict adaptability to other frameworks or face embedding methods, which limits practical applicability across diverse biometric systems.

7. Typos and spelling errors throughout the paper detract from its polish and readability, impacting the professional presentation.

I have reservations about this paper's suitability for ICLR, which typically emphasizes contributions that advance fundamental deep learning theory, introduce novel architectures, or develop techniques with broad applicability and generalizability. In contrast, this paper is an applied study focused on a specific application, which may not align well with ICLR’s core focus areas.


[1] M. Gomez-Barrero and J. Galbally, “Reversing the irreversible: A survey on inverse biometrics,” Computers & Security, vol. 90, p. 101700, Mar. 2020, doi: 10.1016/j.cose.2019.101700.

**Questions:**

See above

---

### Official Review · Reviewer_dPhd · 2024-11-01

**Soundness:** 2
**Presentation:** 2
**Contribution:** 2
**Rating:** 3
**Confidence:** 4

**Summary:**

This work try to exploit Kolmogorov-Arnold Network for conducting embedding-to-face attacks against state-of-the-art face recognition and PPFR systems.

**Strengths:**

The paper is the first approach to use KAN for embedding-to-face task.

**Weaknesses:**

1. The intention of the paper is not clear. Essentially, this article describes a generative task that creates facial images based on the features of facial recognition. Such work cannot be regarded as an attack on facial recognition systems. The authors have not explored issues related to the sources of data for attackers and victims, nor have they discussed the perturbation constraints involved in image attacks.
2. The three embedding-based losses mentioned by the authors in the loss design essentially do the same thing, there is no need to divide them into three separate losses.

**Questions:**

No

---

### Official Review · Reviewer_DBhY · 2024-11-03

**Soundness:** 4
**Presentation:** 3
**Contribution:** 2
**Rating:** 3
**Confidence:** 5

**Summary:**

The paper proposes a new method for face reconstruction from face recognition embedding. Authors proposed a face embedding mapping (FEM) models to learn the distribution mapping relation between the embeddings from the initial domain and target domain. They provide two variants, FEM-KAN and FEM-MLP. They use their method to attack various privacy-preserving face recognition (PPFR) and typical face reconstruction (FR) models.

**Strengths:**

- Authors proposed effective face embedding mapping (FEM) models to project the embeddings from the initial domain and target domain.
- The proposed method is used to attack various privacy-preserving face recognition (PPFR) and typical face reconstruction (FR) models.
- Authors explored the application of their method on various types of face embedding, including complete, partial and protected ones.

**Weaknesses:**

- The paper lacks comparison with previous face reconstruction methods in the literature. Authors mentioned different face reconstruction methods from the literature in introduction and related work sections, but there is no experiment to compare the performance (in terms of ASR) with previous methods. Specially, the experiments include different scenarios (for FR and PPFR), which are interesting, but the performance of the proposed method is not compared with previous face reconstruction methods in experiments.
- The evaluations are performed on a customized dataset (Synth-500) and it is not clear why authors have not used standard face recognition datasets (eg LFW, CFP, AgeDB, etc). Authors are suggested to report their results on standard evaluation datasets.

**Questions:**

- Authors have explored the application of their method on various types of face embedding, including complete, partial and protected ones. This is very interesting experiment. However, it is not very clear for reader how authors have applied their method for each type. Therefore, I suggest authors to clarify this in the paper (or appendix). In addition, is such evaluation for the first time in this paper or if authors followed a previous evaluation in the literature?
- In the introduction, authors mentioned that "current face reconstruction methods focus on face image reconstruction from embeddings of normal FR models". Is there any difference in using the proposed method for attacking FR and PPFR in the proposed method? If there is any difference please clarify. If there is no difference, can previous methods in the literature be also used to attack PPFR (similar to FR models)?
- Three loss functions are used in the proposed method, which are all applied on embeddings and reduce different types of distance. The ablation study in Table 6 does not show significant change in performance. Can authors clarify what is the motivation of using each loss function? Is it necessary to have these three terms?

---

### Note · Authors · 2024-11-14

**Comment:**

I have read and agree with the venue's withdrawal policy on behalf of myself and my co-authors.

**Withdrawal Confirmation:**

I have read and agree with the venue's withdrawal policy on behalf of myself and my co-authors.